# Gradually Excavating External Knowledge for Implicit Complex Question Answering

**Chang Liu[1], Xiaoguang Li[2], Lifeng Shang[2], Xin Jiang[2], Qun Liu[2],**
**Edmund Y. Lam[1], Ngai Wong[1]**
[1]The University of Hong Kong, [2]Huawei Noah's Ark Lab
lcon7@connect.hku.hk
{lixiaoguang11, Shang.Lifeng, Jiang.Xin, qun.liu}@huawei.com
{elam, nwong}@eee.hku.hk

## Abstract

Recently, large language models (LLMs) have gained much attention for the emergence of human-comparable capabilities and huge potential. However, for open-domain implicit question-answering problems, LLMs may not be the ultimate solution due to the reasons of: 1) uncovered or out-of-date domain knowledge, 2) one-shot generation and hence restricted comprehensiveness. To this end, this work proposes a gradual knowledge excavation framework for open-domain complex question answering, where LLMs iteratively and actively acquire external information, and then reason based on acquired historical knowledge. Specifically, during each step of the solving process, the model selects an action to execute, such as querying external knowledge or performing a single logical reasoning step, to gradually progress toward a final answer. Our method can effectively leverage plug-and-play external knowledge and dynamically adjust the strategy for solving complex questions. Evaluated on the StrategyQA dataset, our method achieves 78.17% accuracy with less than 6% parameters of its competitors, setting new SOTA for ~10B-scale LLMs.

## 1 Introduction

Recently, powerful LLMs such as ChatGPT, GPT4 (OpenAI, 2023), PaLM (Anil et al., 2023), LLaMA and its variances (Touvron et al., 2023; Taori et al., 2023), exhibiting human-alike ability in conversation. It is believed the LLMs memorize knowledge in their parameters from the vast pre-training data (Moiseev et al., 2022; Roberts et al., 2020). Nonetheless, they could still fail to solve open-domain implicit complex questions.

In real-world applications, users might ask questions in arbitrary domain that requires specific knowledge, and expect the model to return not only syntactically fluent but also factually correct answers. Beyond open-domain, the questions can also be multi-step and implicit, consisting of multiple sub-questions that cannot be directly identified from the question language, but require logical reasoning to form a problem-solving strategy. Because of the above challenging characteristics, how to answer open-domain implicit complex questions remains an open question.

For example, in the upper part of Fig. 1, an implicit complex question "Did any citizen of San Antonio vote for Boris Johnson" confuses the LLM because there is no direct information about individual voting history. However, the strategy of searching voting history is invalid does not mean the question is unsolvable. On the top right corner of Fig. 1, the question can be decomposed into sub-questions about Boris Johnson and San Antonio, respectively, and a strategy of checking citizenship can be easily unsealed from the background knowledge (marked in red). Following the strategy, the answer is straightforward because US citizens cannot vote in UK elections. Now if the question is re-asked with the hint of 'citizenship contradiction', the LLM can successfully recall its inner knowledge about Boris Johnson and San Antonio, hence correctly answering the question. However, different from existing works that try to design manual prompts (Wei et al., 2022b; Lyu et al., 2023) to serve as the hint in the above example, we want to stress that this approach is not always valid. Because the key 'citizenship contradiction' is not directly linked to the question text, but relies on the gradually increased knowledge during the solving process (e.g., background about Boris Johnson and San Antonio).

Another problem for solving open-domain complex questions with LLMs is the finite pretrained knowledge. In the bottom part of Fig. 1, we rewrite the question with two less well-known entities out of the LLM's knowledge scope, and it fails again due to limited knowledge.

Therefore, in this work, we propose a pipeline

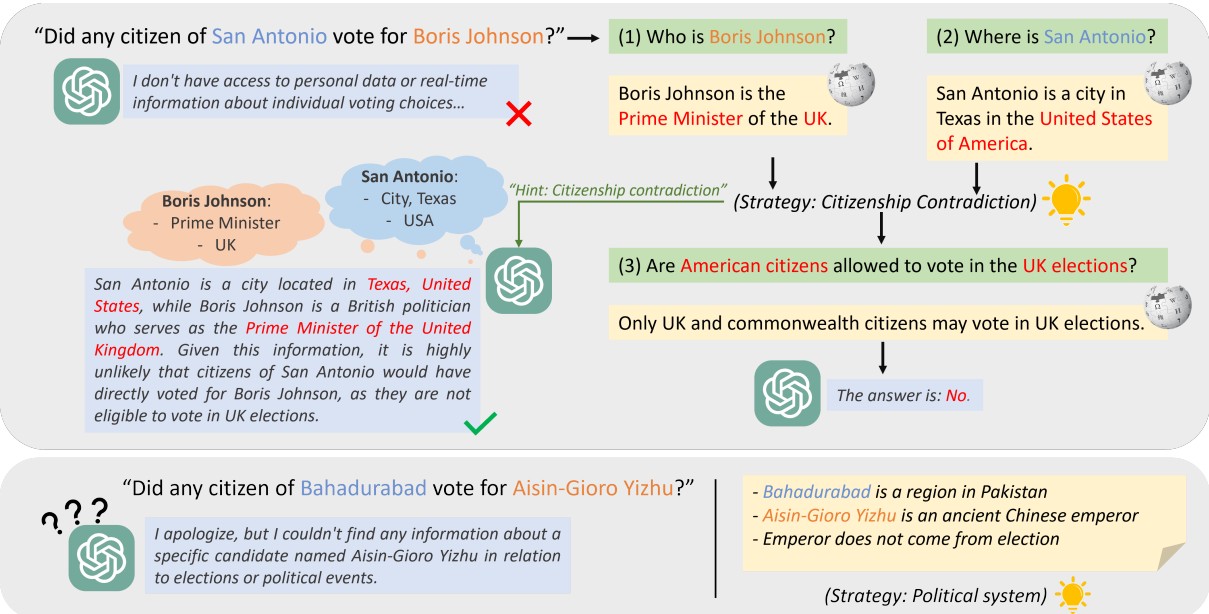

Figure 1: LLMs fail to solve open-domain complex questions due to unrecognized entities and implicit strategies. (1) In the upper part, the LLM fails to answer the question with the one-shot generation, for there is no off-the-shelf answer or evidence to this question. However, the question can be decomposed into several sub-questions and be solved once the citizenship contradiction is identified. If the hint of 'citizenship contradiction' is also given, the LLM can successfully solve the question with the inner knowledge now. (2) But for the bottom case with less well-known entities, the LLM fails again due to a lack of specialized knowledge about 'Aisin-Gioro Yizhu' and hence rejects to answer. Moreover, the strategy of the 'political system' is not likely to be discovered from the question text only, unless enough knowledge is provided. 'Citizenship contradiction' is also a possible solution.

named Gradually Excavating External Knowledge (**GEEK**) to address the main challenges of open-domain implicit complex question answering: external knowledge, multi-step complexity, and implicit logic strategy. Given an open-domain multi-step implicit question, GEEK progressively decomposes the problem into several sub-questions, and iteratively calls different modules for answering the sub-questions. In the end, a final answer is concluded, synthesizing the historical sub-questions and their corresponding answers. Specifically, GEEK consists of three modules, core model, retriever, and extractor. The core model handles logical reasoning and selects an action to perform at each time step, planning the solving strategy purposefully. The retrievers allocate relevant context paragraphs from the external corpus (e.g., Wikipedia) to provide trustworthy knowledge, and the extractor condenses the textual knowledge into brief fact sentences.

During intermediate steps, GEEK can adjust the rest sub-questions based on the gradually increased external knowledge, hence forming a valid strategy like in Fig. 1. Considering there usually exist multiple valid solutions for a question, we enable

GEEK to branch out different sub-questions during the solving process, thus exploring a strategy space and improving the final accuracy.

We verified GEEK on the challenging StrategyQA dataset (Geva et al., 2021), which consists of open-domain multi-step implicit questions. GEEK achieves 78.17% accuracy with less than 6% parameters of its competitors, refreshing the new SOTA for LLMs under ∼300B scale.

Our main contributions are threefold:

- We propose GEEK, a novel pipeline to solve open-domain complex questions by progressively acquiring external knowledge and adjusting its strategy.

- GEEK is able to explore a strategy space to solve the question with different approaches, hence improving the overall performance.

- Our method is evaluated on the challenging StrategyQA benchmark, achieving 77.73% accuracy, surpassing vanilla LLMs such as ChatGPT with 94% parameters less.

## 2 Related Work

***Retrieving external information*** is a widely adopted method that can provide flexible knowledge to extend LLMs for specific-domain tasks. (Izacard and Grave, 2020a) leverages retrievers (e.g., BM25 (Robertson et al., 1995) or DPR (Karpukhin et al., 2020)) to collect relevant passages, and fuse them in the decoder for a final answer. Similarly, (Zhu et al., 2021) proposes AISO, which performs multi-round retrieval with different retriever models of BM25, DPR, and LINK. It then synthesizes the retrievals for a comprehensive conclusion. HopRetriever (Li et al., 2021) adopts a multi-hop manner, which identifies significant entities in previous retrievals, and then uses the entities as queries for next step retrieval.

***Multi-step Implicit Question Answering*** involves complex questions that consist of several single-step questions, whose answers can be directly founded in reference context or inferred via logical deduction. The question is implicit if the decomposition strategy cannot be formulated merely from the question text. StrategyQA (Geva et al., 2021) is a dataset for multi-step implicit question answering, including human-annotated solutions in the form of decomposition questions and corresponding fact sentences from Wikipedia. However, while human beings can achieve 87% accuracy (Geva et al., 2021), the dataset has proved to be very challenging for language models to solve. Merely improving the quality of retrieval (Liang et al., 2022) or decomposition strategy (Katz et al., 2022) cannot effectively boost final answer accuracy.

Several previous studies adopt the design of iterative retrieval and reasoning to solve multi-step complex questions. IRGR (Ribeiro et al., 2022) performs iteratively retrieval to search for suitable premises. ReAct (Yao et al.) lets the model first generate a reasoning sentence about what next action (e.g., search via web API) to be performed, and then execute the selected action. Maieutic Prompting (Jung et al., 2022) introduces a maieutic tree that recursively entails component statements, and uses the concept of 'logical integrity' to verify each step. RR (He et al., 2022) combines CoT (Wei et al., 2022b) with retrieval to verify the correctness of the reasoning process, and achieved an accuracy of 77.73% on the StrategyQA dataset, the SOTA for LLMs below ∼300B scale at that time.

Nonetheless, most of the previous studies using iteratively reasoning for multi-step question answering only focus on datasets such as HotpotQA (Yang et al., 2018), which do not involve implicit questions. Instead, their method assumes a straightforward direction from one step to another. For example, IRCoT (Trivedi et al., 2022) directly uses the 'thought' (intermediate result) from the last step as the query for next-step retrieval, which highly relies on the strong connections between the entities from each step. Different from the above-listed methods, our GEEK modeling the process of purposely excavating external knowledge by composing sub-questions, and formulating a complete strategy gradually during the solving process.

## 3 Problem definition

In this work, we focus on open-domain implicit complex question answering. Given a question $q$, the model is asked to derive the final answer $z$. Specifically, $q$ is open-domain and hence requires some certain background facts to solve, denoted as $\mathcal{F} = \{f_i\}$. The facts come from an external corpus $\mathcal{C}$ (e.g., Wikipedia), not necessarily included in the model's pretraining dataset. The question $q$ is also multi-step, meaning that it can be decomposed into several decomposition questions $\mathcal{D} = \{d_i\}$. Each $d_i$ corresponds to a background fact $f_i$. Noted that $\mathcal{D}$ is usually implicit from $q$, which means some $d_i$ can only be formulated until enough facts $\mathcal{F}_i \subset \mathcal{F}$ is uncovered. Under the GEEK scenario, we define the question state $\mathcal{Q}$ at step $t$ as:

$$\mathcal{Q}_t = (q, \{(d_1, f_1), ..., (d_{t-1}, f_{t-1})\}[, (d_t, f_t)])$$

, where the current decomposition $d_t$ and fact $f_t$ may or may not have been decided yet. $\mathcal{Q}$ includes the question $q$ and historical exploration steps.

## 4 Gradually Excavating External Knowledge

In this section, we introduce GEEK for open-domain complex question answering. GEEK consists of three components, the core model, retriever and extractor model. Iteratively, the core model selects actions to perform from an action space $\mathcal{A}$, conditioned on the current question state $\mathcal{Q}$. Then the selected action is executed and the question state is updated, gradually accumulating external knowledge until a final answer can be drawn.

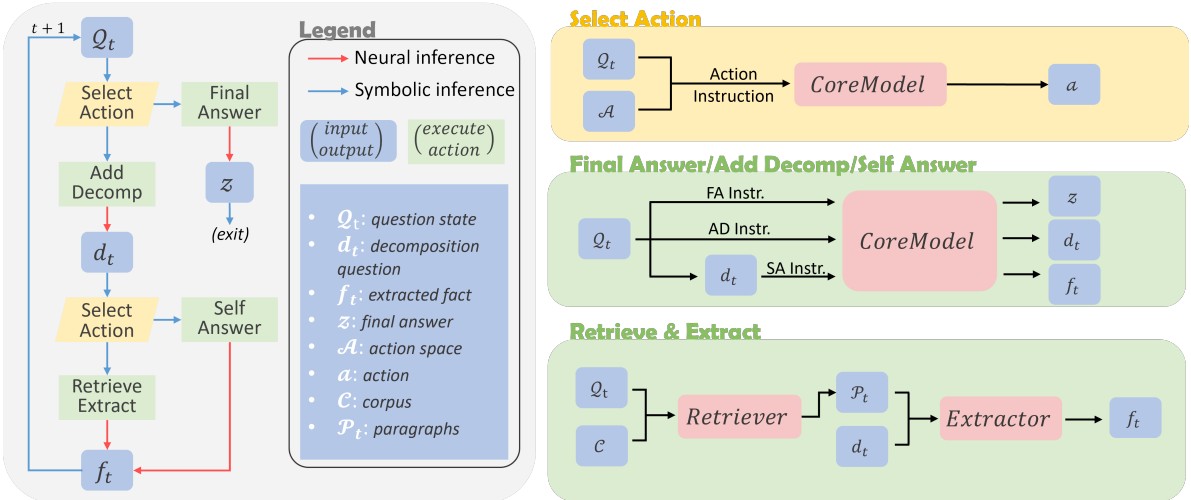

Figure 2: GEEK workflow: the core model, retriever and extractor collaborate to solve complex questions progressively. (Left): In each iteration, based on the question state $\mathcal{Q}_t$, GEEK selects an action and calls the corresponding module to execute. The execution updates the question state in turn, until a final answer $z$ is derived. (Right): The detailed procedure of action selection and execution. For action selection, $\mathcal{Q}_t$ and $\mathcal{A}$ are fed into the core model with the instruction for action selection, and the model outputs an action code $a$. For the execution of Add Decomp, Final Answer, and Self Answer, the core model outputs corresponding responses following different instructions. At last, for Retrieve & Extract, the retriever firstly retrieves several paragraphs $\mathcal{P}$ from the corpus as external knowledge, and then the extractor answers the decomposition question $d_t$ based on $\mathcal{P}$.

## 4.1 Core model

The core model is a pretrained LLM for sequence-to-sequence text generation (e.g., Flan-T5 (Chung et al., 2022)), acting as the controller of GEEK. For each step $t$, the core model chooses an action $a$ to perform, among the action space $\mathcal{A}$ and conditioned on the current question state $\mathcal{Q}_t$:

$$a = Core(Q_t, A)$$

Besides choosing actions, the core model is also used to execute some types of actions, under action-specific prompts, such as generating decomposition questions. Details are in Sec. 4.4.

## 4.2 Retriever

In order to utilize external knowledge, we employ a neural retriever, DPR (Karpukhin et al., 2020), to retrieve paragraphs from a vast volume of context $\mathcal{C}$. Given the decomposition question $d_t$ as query, the retriever returns top $k$ relevant paragraphs:

$$\mathcal{P}_t = \{p_1, p_2, ..., p_k\} = Retr(d_t, \mathcal{C})$$

, where $Retr$ stands for the retriever and each $p_i$ denotes a paragraph. Due to the enormous amount of context in $\mathcal{C}$, full-size retrieval is time costly. For efficiency, we use two nested DPR bi-encoder models (Karpukhin et al., 2020), namely the document retriever and the paragraph retriever. Firstly,

we deploy the title retriever for a document-level retrieval, to shrink the context space to $k_{doc} = 100$ documents, where the context embeddings are built from each document's title and first paragraph. Then the paragraph retriever performs a secondary search on the paragraph level, outputting the top $k$ matched paragraphs among the 100 documents.

## 4.3 Extractor

Even though we retrieve $k$ paragraphs from the vast $\mathcal{C}$, these retrieved paragraphs are still too long to be input into the core model. Therefore, another specialized extractor is used to condense the $k$ paragraphs into concise fact sentences for $d_t$:

$$f_t = Extractor(d_t, \mathcal{P})$$

We use FiD architecture (Izacard and Grave, 2020b) for the extractor. Instead of extracting facts locally from each of the $k$ paragraphs, FiD can perceive all the paragraphs simultaneously and generate more comprehensive results.

## 4.4 GEEK Pipeline and Action Space

As shown in Fig. 2, GEEK iteratively selects and executes actions to solve implicit complex questions. The procedure involves neural inference where neural networks generate text outputs that are less interpretable due to the black-box nature of neural

networks, as well as symbolic inference which follows strict rules. Based on the question state $\mathcal{Q}_t$ at each round, the core model decides whether a final answer can be made or more decomposition needs to be explored. If the latter, it also determines whether external knowledge should be retrieved, or the decomposition can be directly answered using the fact sentences so far. The newly acquired facts are added to the question state for the next iteration. Next, we introduce the details for each action:

- **FinalAnswer** (*core model*) If enough background knowledge is acquired and a final conclusion can be drawn, the core model should output the final answer to $q$. For the real implementation, we let the model summarize the facts from previous steps as a self-CoT, and then conclude the final answer (yes or no).

- **AddDecomp** (*core model*) Based on current $\mathcal{Q}_t$, the core model generate a next-step decomposition question $d_t$. The generation is conditioned on previous decompositions and facts. Hence the model could adjust the reasoning strategy on-fly, and benefit from the gradually enhanced external knowledge.

  To further improve the comprehensiveness of the strategy and avoid generating unsolvable decomposition questions, we design a *pre-answer* trick. Instead of generating $d_t$ only, we use an explicit prompt to instruct the model to generate all the remaining decompositions with corresponding pseudo answers: $(d_t, \tilde{f}_t, d_{t+1}, \tilde{f}_{t+1}, \cdots)$. Hence, the strategy leading by $d_t$ could be more coherent and solvable. Note that the generated pre-answers are not necessarily correct, and all the redundant generations except $d_t$ only serve as generation auxiliaries, which will be removed before adding to the question state $\mathcal{Q}_t$.

- **Retrieve & Extract** (*retriever*) Once a new decomposition is added, the core model would decide whether this decomposition question needs external knowledge to answer. If yes, this action is executed and the retriever is called to retrieve the top $k$ most relevant paragraphs $\mathcal{P}_t$ corresponding to the current decomposition $d_t$. Once $\mathcal{P}_t$ is retrieved, the extractor model would read the $k$ paragraphs and generate a concise fact $f_t$ as the answer to $d_t$.

  For the extractor, we also use the generated pseudo answer $\tilde{f}_t$ from the 'AddDecomp' action as a reference. We formulate the input to extractor with the prompt: 'Answer the question $d_t$ based on the context $p_i^{(t)}$, a reference but not necessarily correct answer is $\tilde{f}_t$'. Therefore, the extractor knows what kind of information should be extracted among possibly multiple aspects, avoiding correct but not relevant answers (e.g., for decomposition 'Who is xxx?', possible aspects include nationality, career, education, family, etc.).

- **SelfAnswer** (*core model*) For some decomposition questions that are pure logical deduction, or the required knowledge has been included in the question state already, no external evidence is needed and hence retriever and extractor are not used. In this case, a self-answer prompt is used and the core model would answer $d_t$ directly, outputting $f_t$.

## 4.5 Strategy Exploration

Considering that there often exist multiple possible strategies to solve an identical question, we also extend the GEEK to explore a strategy space for different solutions. Specifically, in the step of *AddDecomp*, the core model can return multiple decomposition questions using beam search. For each different decomposition $d_t^{(i)}$, a copy of current question state $\mathcal{Q}_t$ is created, and updated by the new decomposition, forming $\mathcal{Q}_t^{(i)}$. Then the copies carry on for the rest solving process independently.

We emphasize that our method Strategy Exploration (SE) is different from Self-Consistency (Wang et al., 2022), which outputs multiple CoT solutions with the one-shot generation, and then takes the majority. Under the scenario of GEEK with SE, the question branches into $n = 4$ different strategies at every iteration, hence formulating a latent solution tree. The diverged decomposition would lead to different retrieval results and generated facts, hence is an exploration of the strategy space. Due to computation constraints, we limit the expansion number to be at most 16 (i.e., expansion rate $n = 4$ at each decomposition and for at most 2 iterations). The majority vote is used to derive the final answer.

## 5 Case Study

In Fig. 3 we show a full inference process of GEEK. The inference involves in total three iterations. In

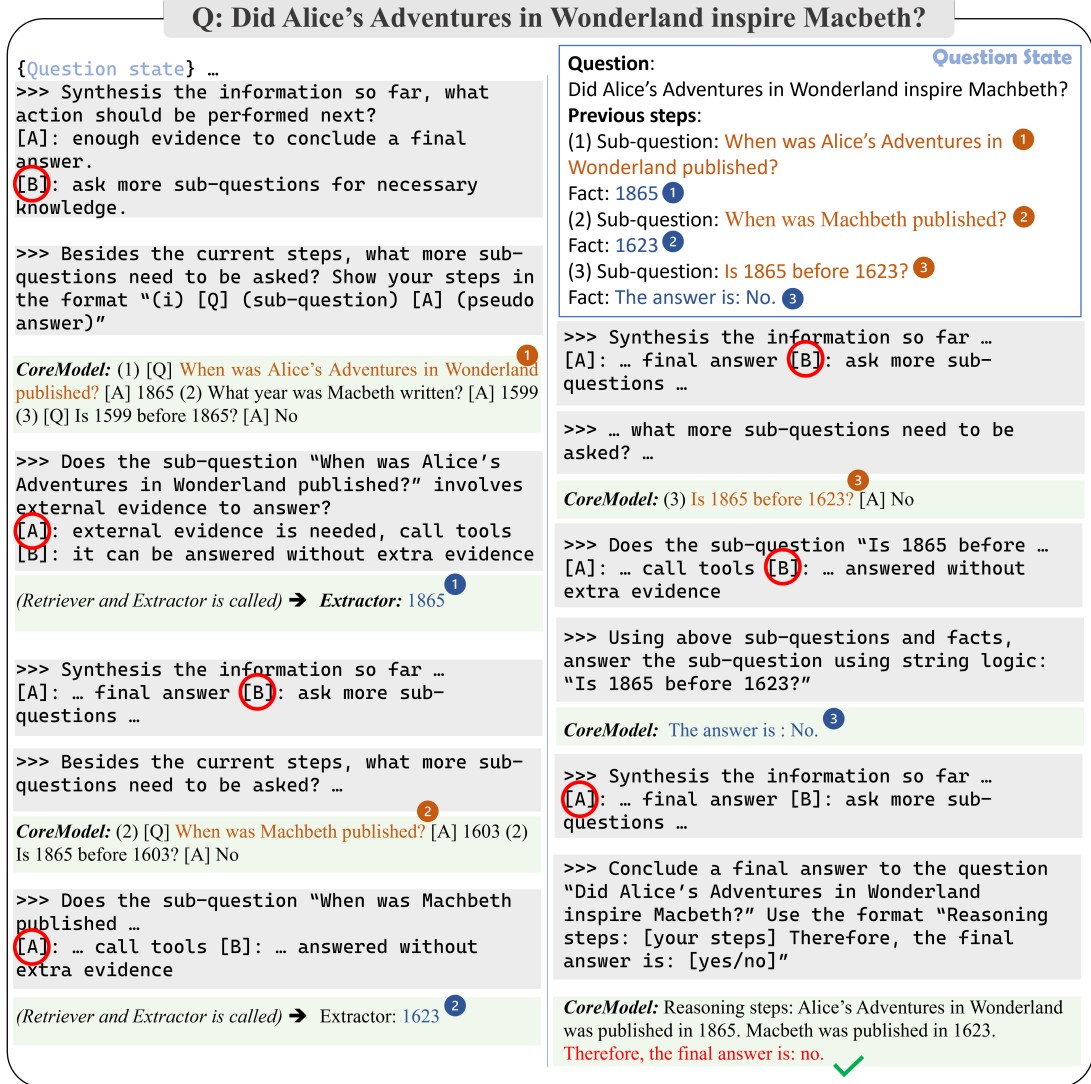

Figure 3: Full process of GEEK inference. For each round, the prompts are shown in gray, and the current question state is also given to the model as input. Model responses are shown in green and action selection is represented by a red circle to save space. On the top right corner, the question state is listed, where the historical states of sub-question and fact are gradually added during the inference (best viewed in color and numerical marks).

order to save space, we omit the input question state for each step, but show it in the top right corner. Marked in colored numbers, the question state is gradually enriched as the decomposition questions and corresponding facts are added. Also, for each action selection step, we circle out the core model's choice. Some repeated prompts are also abbreviated, due to the page limitation.

As can be observed in the example, the GEEK decomposes the original question into three sub-questions, and takes the strategy of 'temporal confliction' to solve the problem. For sub-questions '(1)' and '(2)', external knowledge is retrieved and the original pseudo answers are verified and corrected (e.g., '1599' → '1623'). The strategy is

also dynamically adjusted as more facts are acquired (e.g., 'Is 1599 before 1865?' → 'Is 1865 before 1623?'). For the last logical sub-question that requires no external knowledge, the core model chooses to self-answer it without calling retriever and extractor. When there is enough information, the final answer 'no' is given with reasons.

## 6 Experiment

In this section, we compare GEEK with previous baselines and conduct ablation studies to analyze the contributions of specific modules.

### 6.1 Dataset and Preprocessing

We use the StrategyQA dataset (Geva et al., 2021) to evaluate our method. The dataset consists of

| Method | Backbone | Retrieve | Specification | SQA |
|---|---|---|---|---|
| ChatGPT (Qin et al., 2023) | GPT-3.5 (175B) | ✗ | Without CoT | 59.2 |
| ChatGPT (Qin et al., 2023) | GPT-3.5 (175B) | ✗ | CoT | 62.5 |
| FaithfulCoT (Lyu et al., 2023) | code-davinci-002 (175B) | ✗ | - | 73.2 |
| (Xie et al., 2023) | code-davinci-002 (175B) | ✗ | - | 77.2 |
| (Lazaridou et al., 2022) | Gopher (280B) | ✓ | - | 66.2 |
| Visconde (Pereira et al., 2023) | text-davinci-002 (175B) | ✓ | CoT | 69.43 |
| RR (He et al., 2022) | text-davinci-002 (175B) | ✓ | CoT | *77.73* |
| PaLM (Chowdhery et al., 2022) | PaLM (540B) | ✗ | - | 73.9 |
| PaLM (Anil et al., 2023) | PaLM (540B) | ✗ | CoT + SC | 81.6 |
| PaLM2 (Anil et al., 2023) | PaLM2 (340B) | ✗ | - | **90.4** |
| GEEK (ours) | Flan-T5 (11B) | ✓ | CoT | 75.98 |
| GEEK (ours) | Flan-T5 (11B) | ✓ | CoT+SE | **78.17** |

Table 1: Experiment results on strategyQA dataset. GEEK achieves the SOTA accuracy for LLMs in ∼10B scale, and surpasses all the previous methods with backbone under 300B scale, using only 6% parameters or less.

2061 samples in the train set and 229 in the dev set. Another 490 samples without labels are provided for the test set, and results for that can be uploaded to their website for evaluation [1]. For train and dev samples, each question is provided a human-annotated strategy $\mathcal{D}$ in the form of decomposition questions. Golden supporting paragraphs $p$ for each $d$ are also given, which are from 36.6M Wikipedia processed corpus. In addition, human-annotated background facts $\mathcal{F}$ are also provided.

Nonetheless, the provided background facts are not strictly mapped with the decomposition, either in the aspect of total number or sequential order. Additionally, over 25% human annotated decomposition questions refer to previous $i$-th decomposition by symbol '#i'. We find that these symbols cannot be simply filled by decomposition's corresponding answer, because some '#i' may refer to an entity in the decomposition question text. For the above two reasons, we use GPT4 (OpenAI, 2023) to refine the annotations. We provide the question together with the final answer, golden facts and decomposition questions to GPT4, and prompt it to fill the '#'s in annotations. Meanwhile, we also ask it to give a concise answer for each decomposition, according to the golden facts and the final answer. Unless specified, all the experiment results below are from the GPT4 processed version.

### 6.2 Detailed Settings

For retriever, vanilla DPR (Karpukhin et al., 2020) is used with BERT-base-uncased (Devlin et al.,

[1]https://allenai.org/data/strategyqa

2018) as backbone. By default, $k = 10$ paragraphs will be retrieved during GEEK inference. The extractor is a FiD (Izacard and Grave, 2020b) with Flan-T5-3B as the backbone. During training, the extractor is fed golden paragraphs and remaining retrieval paragraphs, to satisfy the fixed number $k$.

For the core model, we adopt Flan-T5-11B (Chung et al., 2022; Raffel et al., 2020). The model is trained for multiple tasks including action selection and execution of three actions, as described in Fig. 2. The tasks are trained in parallel, with input-output pairs built from human annotations and in the same instruction format as Fig. 3. We train the model on 8 V100 GPUs. Due to the LLMs' out-of-memory problems, the deepspeed flatform is used (Rasley et al., 2020). During inference, we utilize the accelerate package with offloading for acceleration (Sylvain et al., 2022).

### 6.3 Comparison with other Baselines

In this section, we compare the proposed GEEK with other baselines on the strategyQA dataset. As shown in Tab. 1, GEEK yields 78.17% accuracy on the StrategyQA dataset, with a much smaller model size (11B) than previous baselines. Our method sets a new SOTA for the ∼10B LLMs, and also is the second best among all existing methods except PaLM. Although GEEK is finetuned with supervision, the accuracy of GEEK is still considerably high considering its size in comparison with other baselines. It is also worth noting that due to the indirection of using LLM APIs, finetuning or adaption for special domain tasks is not easy. For

|          | De | RE | SE | Acc   |
|----------|----|----|----|-------|
| Zero-shot | ✗  | ✗  | ✗  | 62.01 |
| CoT       | ✗  | ✗  | ✗  | 70.74 |
| +De       | ✓  | ✗  | ✗  | 71.50 |
| +RE       | ✓  | ✓  | ✗  | 75.98 |
| Full      | ✓  | ✓  | ✓  | 78.17 |

Table 2: Ablation study results. The three columns denote whether an action is performed under GEEK. ('De': 'AddDecomp', 'RE': 'Retrieve and Extract', 'SE': 'Strategy Exploration')

|         | Human  | Equal  | GEEK   |
|---------|--------|--------|--------|
| ChatGPT | 13.54% | 24.02% | 62.45% |

Table 3: ChatGPT Assessment of prediction results.

example, only Visconde (Pereira et al., 2023) and RR (He et al., 2022) successfully leverage external knowledge, but with a relatively less capable backbone text-davinci-002. However, the external knowledge is proved to benefit the task effectively, and RR achieves SOTA performance at its time.

Specifically, while the initially finetuned GEEK yields 75.98% accuracy, by using SE, the accuracy is improved significantly, to 78.17%. This process requires no retraining yet can boost final accuracy, where GEEK explores the strategy space and tries to solve the question via multiple paths.

### 6.4 Ablation Study

We also analyze the contribution of the different components in GEEK. Results are shown in Tab. 2. The line of 'CoT' denotes that the core model directly answers the question as in the 'FinalAnswer' action, following the CoT approach (Wei et al., 2022a) without iterative reasoning and knowledge retrieval. After finetuning, the accuracy is 70.74% accuracy, 8.73% higher than zero-shot Flan-T5 but 7.73% lower than the full version of GEEK. We find that performing the action 'Retrieve&Extract' could efficiently increase the accuracy to 75.98%, justifying the motivation of leveraging external knowledge for solving open-domain questions.

### 6.5 ChatGPT Assessment

To further evaluate the quality of the strategy generated by GEEK, we also leverage the GPT4 to simulate a human assessment. Specifically, we show it the human-annotated decomposition questions and corresponding facts, as well as the GEEK-

generated decomposition-fact pairs, and let the model choose which is more informative and faithfully correct. The results are listed in Tab. 3. It turns out that 62.45% of the GEET-generated decomposition-fact pairs are preferred.

## 7 Conclusion

We present GEEK, a pipeline to progressively excavate external knowledge for boosting LLM's capability in solving open-domain multi-step implicit questions. Interactively, GEEK decomposes the question into explicit sub-questions to retrieve external knowledge, and the accumulated knowledge enlightens the model to form a better strategy in turn. The proposed method also provides explainability by showing the full reasoning process, supported by retrieved evidence. By using SE, GEEK can also explore the strategy space and try different approaches to solve the complex question, which increases the performance further. Experiment results justify our design. With GEEK, 78.17% accuracy is achieved using less than 6% size of the competitors, refreshing the SOTA accuracy for ∼10B LLMs. As an alternative to the paradigm of scaling for larger models and more pretraining data, we hope this research could inspire more future works to investigate how to organically excavate external knowledge and progressively formulate strategies for solving open-domain implicit questions.

## Limitations

Albeit achieving outstanding overall accuracy with a significantly smaller model size, the GEEK is not without limitations. First of all, as long as neural reasoning is involved, the hallucination problem is inevitable in theory, due to the black-box nature of neural networks. We want to stress that by using a retriever and an extractor, the hallucination problem can be alleviated from factual references, but not completely avoided. Secondly, the logic of GEEK is not guaranteed to be correct. It is possible that GEEK gives the correct answer but wrong solving steps, or correct intermediate steps but wrong final answer. Lastly, due to the scarcity of public datasets like StrategyQA for open-domain complex question answering, it is difficult for us to research this problem under more datasets and different task settings. We expect future works that may come up with more suitable datasets like StrategyQA.

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

# Appendix

## A  Training Details

For training the core model, we use the refined annotation of StrategyQA dataset. The original StrategyQA dataset provides human-annotated decomposition questions, fact sentences, and supporting paragraphs from Wikipedia. The decomposition questions break the original open-domain complex question into several simple and explicit sub-questions. The fact sentences contain evidence and background knowledge for solving the main question. The supporting paragraphs are where the fact sentences are grounded, and are one-to-one matched with the corresponding fact sentences. Nonetheless, the fact sentences are not one-to-one mapped to each decomposition question. Therefore, we use GPT4 to refine the annotation, extracting a concise answer for each decomposition question from the set of human-annotated fact sentences. Explicit instruction is given to GPT4 to force the model to use information that is faithful to the human-annotated fact sentences only.

After that, the core model is trained via supervised fine-tuning. Specifically, we use ground-truth annotations (decomposition, paragraphs, and fact sentences) to build a full-solving process for each question. The ground-truth action to be performed also depends on the process. For example, if there are still more ground-truth decomposition questions, the model should select 'AddDecomp'. And if all decomposition questions have been visited, the model should select 'FinalAnswer'. By doing so, we build multiple input-output pairs simulating each timestep of the solving process, and use these pairs to train the core model in parallel.

## B  More Comparison with Other Baselines

Besides the results shown in Tab. 1, we also list more comparison results with other baseline methods, whose backbone models are similar in size to ours. As shown in Tab. 4, our method significantly surpass the other baselines, with improvement larger than 7.62%.

## C  Results with Other Backbones

We also implement GEEK with other backbone models, to verify the generality of our method. In this section, we select the similar-scale LLaMA models for comparison. As shown in Tab. 5, GEEK performs well with all the backbone models. The LLaMA-13B backbone yields similar accuracy as Flan-T5-11B, demonstrating the generality of our GEEK pipeline. However, LLaMA is trained without instruction tuning, and both LLaMA-7B and LLaMA-13B perform slightly worse than Flan-T5-11B. This observation suggests that instruction tuning is helpful for the task of StrategyQA, but a larger model (e.g., LLaMA-13b) can reduce the gap.

## D  Prompt Examples

In this section, we show all the prompts used in GEEK, please also refer to Sec. 4.4 for more details of the actions. The terms such as $\{Question\_state\}$ represents the placeholder to be substitute by corresponding text values.

- System Prompt (at the beginning of every input):

```
"Try_to_solve_a_multi-step_
    open-domain_question._{Question_state}"
```

- ActionSelection Prompt1 (begin of each round):

```
"Synthesis_the_information_so_
    far,_what_action_should_be
    _performed_the_next?_\n[A
    ]:_enough_evidence_to_
    conclude_a_final_answer._\
    n[B]:_ask_more_sub-
    questions_for_necessary_
    knowledge."
```

- If final answer (action [A]):

```
"Conclude_a_final_answer_to_
    the_question_{Q}._Use_the_
    format_\"Reasoning_steps:_
    [your_reasoning_steps]_
    Therefore,_the_final_
    answer_is:_[yes/no]\""
```

- If add decomp (action [B]):

```
"Besides_the_current_steps,_
    what_more_sub-questions_
    need_to_be_asked?_Show_
    your_steps_in_the_format_
    \"(i)_[Q]_(sub-question)_[
    A]_(pseudo_answer)\""
```

| Method | Model size | Acc |
|---|---|---|
| UL2 (Qin et al., 2023) | 20B | 59.0 |
| StableVicuna INT8 (Hartill et al., 2023) | 13B | 61.7 |
| GR+RATD (Hartill et al., 2023) | 440M | 64.2 |
| KARD (Kang et al., 2023) | 3B | 70.55 |
| GEEK (ours) | 11B | **78.17** |

Table 4: More comparison with other baselines.

| LLM | Acc |
|---|---|
| LLaMA-7B | 74.67 |
| LLaMA-13B | 77.73 |
| Flan-T5-11B | 78.17 |

Table 5: More backbone results.

- ActionSelection Prompt2 (when new decomp is added):

```
"Does␣the␣sub-question␣{Decomp
    }␣involves␣external␣
    evidence␣to␣answer?␣\n[A]:
    ␣external␣evidence␣is␣
    needed,␣call␣tools.␣\n[B]:
    ␣it␣can␣be␣safely␣answered
    ␣by␣logical␣inference␣
    without␣extra␣evidence"
```

- If call tools (action [A]):

```
# retriever input
"Question:␣{Q},␣Sub-question:␣
    {Decomp}"

# extractor input
"Based␣on␣the␣following␣
    context,␣answer␣the␣
    question:␣\"{decomp}\"␣(A␣
    reference␣but␣not␣
    necessarily␣correct␣answer
    ␣is:␣\"{pseudo_answer}\")
    Context:␣{paragraph}"
```

- If self answer (action [B]):

```
"Based␣on␣the␣sub-questions␣
    and␣facts,␣use␣strict␣
    logic␣to␣answer␣the␣sub-
    question:␣{Decomp}"
```

## E  Error Analysis of GEEK

We manually analyzed all the error samples in the dev set. The reasons for wrong predictions are categorized into 4 types listed below (with actual examples):

- **Unreasonable decomposition**: the predicted decomposition does not effectively lead to a valid solution to the original answer. For example:

  – *Question*: "Would the author of Little Women have remembered the ratification of the 13th Amendment?"
  – *Ground truth*: "(1) When was the 13th Amendment ratified? (2) Who wrote Little Women? [Louisa May Alcott] (3) What years was Louisa May Alcott alive? [1832-1888] (4) Did the ratification of the 13th Amendment occur sometime during 1832-1888?" [Final answer: yes]
  – *Prediction*: "(1) When was the 13th Amendment ratified? (2) When was Louisa May Alcott born? [1832] (3) Is 1865 before 1832?" [Final answer: no]

- **Wrong action selection**: the model selects the wrong action to be performed (e.g., conclude final answer too early, incorrectly call retriever, or attempt self-answer)

- **Incorrect facts**: Retriever and extractor output incorrect facts (either irrelevant or factually incorrect). For example:

  – *Decomposition question*: "What are the numbers that are used in the scoring system in table tennis?"
  – *GT*: "11 and 21" (for old rules).
  – *Pred*: "15 points, 30 points, and 40 points".

- **Logical deduction error**: the model gives the wrong final answer from correct decompositions and facts. For example:

  - *Question*: "Can the Swiss Guard fill the Virginia General Assembly chairs?"
  - *Facts*: "There are 140 seats in the Virginia General Assembly. The Swiss Guard has a total of 134 members."
  - *GT answer*: "No"
  - *Pred answer*: "Yes"

Based on the above categories, we also statistic the proportion of errors. The results are shown in Tab. 6. Most of the errors are due to bad decomposition generation and incorrect facts. Because of the nature of implicit QA, generating good decomposition questions is challenging. By observing the error cases, we find that logical reasoning and background knowledge are required and critical to generate high-quality decompositions. Therefore, we hypothesize that GEEK could benefit from a larger backbone LLM, which is believed to have more knowledge absorbed and higher reasoning ability. This is also partially verified by the results in Tab. 1 and Tab. 5.

For the factual errors in generated facts, as also mentioned in the section 'Limitation', the extractor's neural processing mechanism makes the hallucination problem inevitable. Based on the observation of error cases, we find that two main reasons result in incorrect facts: (1) the retriever fails to find relevant paragraphs corresponding to the decomposition question, and (2) the extractor outputs summarized fact sentences that are not faithful to the paragraph. For (1), a more powerful retriever (e.g., search engine) or more affluent corpus beyond Wikipedia could help. For (2), faithful QA techniques, such as changing the generative task into an extractive task, might be a remedy. However, constraints by the computation resources, and also considering that this research mainly focuses on implicit question answering, we leave this problem for future works.

| Error type | Percentage |
|---|---|
| Unreasonable Decomposition | 40% |
| Wrong action selection | 8% |
| Incorrect facts | 54% |
| Logical deduction error | 20% |

Table 6: Error cases and proportion.