# OpenReview forum: "Gradually Excavating External Knowledge for Implicit Complex Question Answering"
_EMNLP/2023/Conference — EMNLP 2023 Findings_

### Official Review · Reviewer_4HSn · 2023-08-04

**Soundness:** 3

**Excitement:**

3: Ambivalent: It has merits (e.g., it reports state-of-the-art results, the idea is nice), but there are key weaknesses (e.g., it describes incremental work), and it can significantly benefit from another round of revision. However, I won't object to accepting it if my co-reviewers champion it.

**Paper Topic And Main Contributions:**

This work proposes a gradual knowledge excavation framework called GEEK for open-domain complex question answering, where LLMs iteratively and actively acquire extrinsic information, then reason based on acquired historical knowledge. GEEK consists of three modules, core model, retriever and extractor. The core model handles logical reasoning and selects an action to perform at each time step, planning the solving strategy purposefully. The retrievers allocate relevant context paragraphs from the external corpus (e.g., Wikipedia) to provide trustworthy knowledge, and the extractor condenses the textual knowledge into brief fact sentences. The experimental results on the challenging StrategyQA benchmark surpass vanilla LLMs such as ChatGPT with 94% parameters less, refreshing the new SOTA for LLMs under ∼300B scale.

**Questions For The Authors:**

A.Have the authors considered conducting some generalization experiments on other tasks? For example, the CommonsenseQA, this will demonstrate the superiority of the model.
B.Can the authors conduct some error analysis?
C.It seems necessary to present a comparison with models of the same or smaller scale. Why didn't the author do this?

**Reasons To Accept:**

A. This paper carefully designed a pipeline model that gradually excavates knowledge from an external knowledge base to solve complex problems. It solves the problem of one-time retrieval by decomposing problems and iteratively retrieval.
B. This paper designed a rigorous and complex reasoning process, utilizing the existing excellent retrieval algorithm (DPR) and answer extractor (FID) as auxiliary tools to obtain the retrieved facts. By fine-tuning Flan-t5-11B to control the next action selection, decompose complex problems into sub problems, and generate answers through internal knowledge directly. It solve the hallucination problem of LLMs to some extent.

**Reasons To Reject:**

A. It is not sufficient to conduct experiments solely on StrategiyQA, which reduces the applicability and makes it hard to prove the effectiveness of the model.
B. The paper lacks necessary error analysis.
C. The author did not provide a detailed description of how to fine tune the core model for action selection, and how to determine whether the model can do self-answer.
D. The author used GPT4 to re-annotate the data, which increased costs and appeared to be an unfair comparison.

**Reproducibility:**

4: Could mostly reproduce the results, but there may be some variation because of sample variance or minor variations in their interpretation of the protocol or method.

**Reviewer Confidence:**

4: Quite sure. I tried to check the important points carefully. It's unlikely, though conceivable, that I missed something that should affect my ratings.

**Typos Grammar Style And Presentation Improvements:**

Line 192, The formula is difficult to understand
Line 344 'GEED' should be 'GEEK'
Line 210, reference error
Lines 679-680, no appendix exists, should be deleted

---

### Official Review · Reviewer_2gMb · 2023-08-04

**Soundness:** 3

**Excitement:**

3: Ambivalent: It has merits (e.g., it reports state-of-the-art results, the idea is nice), but there are key weaknesses (e.g., it describes incremental work), and it can significantly benefit from another round of revision. However, I won't object to accepting it if my co-reviewers champion it.

**Paper Topic And Main Contributions:**

This paper proposes a pipeline for implicit question answering in the open domain. The pipeline is built upon the core module of LLM, which serves as the decision-making center to determine and execute actions, decompose questions, and gradually obtain the final answer. The authors conducted experiments on the strategyQA dataset and demonstrated the effectiveness of this pipeline.

**Reasons To Accept:**

- The pipeline proposed by the authors for the task of implicit question answering appears to be relatively straightforward to implement and shows promising results when incorporating external knowledge.
- The method description is detailed and the writing is clear.

**Reasons To Reject:**

I believe this work is of high value, as it tackles the specific task of complex problem reasoning. However, I have some concerns:

-  Currently, tools like langchain, which integrates ReaAct, can already build tool learning pipelines by calling interfaces. These models have shown promising results on complex tasks that require external knowledge. Considering the availability of such tools, what are the advantages of this research?
- The experiments in this paper were only conducted on the strategyQA dataset, including the main experiment and the ablation stud of CoT/SE. There is a lack of in-depth analysis of the reasons for errors and error statistics, making it difficult to assess the actual effectiveness of the pipeline.
- There is a lack of actual examples and prompt demonstrations.

**Reproducibility:**

4: Could mostly reproduce the results, but there may be some variation because of sample variance or minor variations in their interpretation of the protocol or method.

**Reviewer Confidence:**

5: Positive that my evaluation is correct. I read the paper very carefully and I am very familiar with related work.

---

### Official Review · Reviewer_DMUD · 2023-08-12

**Soundness:** 3

**Excitement:**

3: Ambivalent: It has merits (e.g., it reports state-of-the-art results, the idea is nice), but there are key weaknesses (e.g., it describes incremental work), and it can significantly benefit from another round of revision. However, I won't object to accepting it if my co-reviewers champion it.

**Missing References:**

1. Trivedi, Harsh, et al. "Interleaving retrieval with chain-of-thought reasoning for knowledge-intensive multi-step questions." arXiv preprint arXiv:2212.10509 (2022). A similar work. Please compare with this paper, this paper also achieves high performance with small models using similar pipelines proposed in the submission.

**Paper Topic And Main Contributions:**

The authors propose to extract knowledge bases step by step to solve the open-domain complex question-answering task instead of extracting all knowledge context at one time. They decompose questions into sub-questions for extracting knowledge. In addition, they use FLAN-T5 as a transaction liked model to determine when to stop extracting knowledge using sub-questions and answer questions. The experiments are only conducted on StrategyQA and achieve 78.17% accuracy compared to the previous SOTA performance, which is 77.73%. But the parameter size of the proposed model is 6% of the previous SOTA model.

**Reasons To Accept:**

1. The model achieves the SOTA performance with a much smaller parameter size.
2. The authors propose a gradual knowledge excavation framework for open-domain complex question answering.
3. The authors announce they are the first to propose a novel pipeline to solve open-domain complex questions by progressively acquiring external knowledge and adjusting its strategy.

**Reasons To Reject:**

1. Readability. Many typos and Method related parts such as Sections 4 and 5 can be improved for easier understanding.
2. More experiments on different open domain question answering datasets and more relevant model comparisons are needed.

**Reproducibility:**

3: Could reproduce the results with some difficulty. The settings of parameters are underspecified or subjectively determined; the training/evaluation data are not widely available.

**Reviewer Confidence:**

4: Quite sure. I tried to check the important points carefully. It's unlikely, though conceivable, that I missed something that should affect my ratings.

**Typos Grammar Style And Presentation Improvements:**

Typos:
Page.2 LINE 100, right: 77.73% -->78.17%
Page.3 LINE 188, right: eternal --> external
Page.3 LINE 195, right: q is open-main -->q is open-domain
Page.4 LINE 210, left: Details in Sec. ??
There are many other typos, a careful proof reading is suggested.

---

### Official Review · Reviewer_Ab27 · 2023-08-14

**Soundness:** 3

**Excitement:**

3: Ambivalent: It has merits (e.g., it reports state-of-the-art results, the idea is nice), but there are key weaknesses (e.g., it describes incremental work), and it can significantly benefit from another round of revision. However, I won't object to accepting it if my co-reviewers champion it.

**Missing References:**

[1] Yao, Shunyu, Dian Yu, Jeffrey Zhao, Izhak Shafran, Thomas L. Griffiths, Yuan Cao, and Karthik Narasimhan. "Tree of thoughts: Deliberate problem solving with large language models." arXiv preprint arXiv:2305.10601 (2023).

**Paper Topic And Main Contributions:**

The paper introduces a framework for open-domain implicit question answering that requires multi-step reasoning and access to external knowledge.
Motivated by LLMs’ limitations in “uncovered or out-of-date parametric knowledge”, the proposed approach aims to progressively break down complex problems and retrieve/extract external knowledge to answer implicit complex questions.

The main contributions include: (1) a Gradually Excavating External Knowledge (GEEK) framework that uses three modules—core, retriever and extractor—to perform multi-step reasoning with access to tools. The core module determines question decomposition and the halting condition. The retriever module performs retriever at the title and paragraph level, and the extractor module condenses retrieved content into concise sentences. (2) Employing parallel strategies to solve a problem by simultaneously exploring multiple decomposition trajectories. (3) A pre-answer trick that uses mock answers output from the model to constrain the pipeline search. (4) Evaluation on the StrategyQA dataset demonstrate competitive performance 78.17% accuracy using a much smaller model (FLAN-T4-11B). (5) Ablation studies demonstrate the effectiveness of knowledge retrieval.

**Questions For The Authors:**

Q1. How the core module is fine-tuned exactly to decompose question and decide when a final answer is reached?

Q2. What are the technical challenges in implementing the core module and how are they addresses if any?

Q3. How does multi-step iterative decomposition improves model performance? What is the baseline performance if the model only performs a single retrieval step/reasoning?

Q4. The proposed approach is tailored for implicit QA; is it also applicable to hoptopQA?

Q5. What's the role of GPT4 dataset refinement on model performance? What's the baseline performance without GPT4 refinement? Is there any plan to release the refined dataset?

Q6. How does parallel "strategy exploration" differ from tree-of-thought [1].

**Reasons To Accept:**

1. The paper is well-motivated and the use case demonstrations for implicit question answering is interesting and insightful.

2. The paper proposes a comprehensive multi-step question answering framework. The three main components of the system are well-designed: action space of the core module can decide decomposition/halting/self-answer, retriever and extractor can retrieve and condense external knowledge, parallel reasoning paths can improve the quality/robustness of the results.

3. Ablation studies validate the effectiveness question decomposition and knowledge retrieval components.

**Reasons To Reject:**

Overall, I liked the idea and proposed framework. My main concern is in the lack of technical detail, sufficient experimental validation, and potential challenges with reproducing the results.

1. *Lack of technical detail in how the core module is fine-tuned.* The proposed approach is mostly described at a conceptual level and lacks in-depth technical detail regarding how the core module is fine-tuned. This is of particular concern given the sophistication of the proposed system and the importance of the core module in performing question decomposition, output final answer, and deciding tool usage. The absence of such technical details could hinder the reproducibility of the approach, especially since the source code is not available.

2. *Insufficient validation of iterative problem decomposition.* In the related work, the paper argues the main contribution over other retrieval-based multi-step QA methods is on the application of implicit questions and the need for multi-step progressive reasoning or tool use. Although ablation study confirms the effectiveness of question decomposition in general, it lacks analysis on the number of iterative steps performed and how it improves model performance. As the model has only been validated on a single dataset, StrategyQA, it remains unclear how the proposed approach would fare on “simpler” cases such as hotpotQA.

3. *GPT4 dataset refinement confounds the contribution of this paper.* The proposed approach is fine-tuned on a version of the StrategyQA dataset that is refined by GPT4. This raises the possibility that the fine-tuned model may have learned or distilled behaviors from GPT4, behaviors that might have been difficult to acquire from the original dataset. While distillation is an accepted technical method, the paper does not clarify the degree to which GPT4 refinement contributes to improving model performance. Consequently, this creates an unfair comparison with other models that are not fine-tuned on GPT4 outputs and depend on prompting.



**Reproducibility:**

2: Would be hard pressed to reproduce the results. The contribution depends on data that are simply not available outside the author's institution or consortium; not enough details are provided.

**Reviewer Confidence:**

3: Pretty sure, but there's a chance I missed something. Although I have a good feel for this area in general, I did not carefully check the paper's details, e.g., the math, experimental design, or novelty.

---

### Meta-Review · Area_Chair_KUxR · 2023-09-20

**Recommendation:** 4

**Metareview:**

The paper introduces GEEK, a method that employs multi-step reasoning and retrieval from external knowledge sources to answer implicit questions. Such questions typically pose challenges for LLMs due to outdated parametric knowledge. GEEK incorporates both question decomposition and a halting condition. On StrategyQA, GEEK achieves a new state of the art of 78.17%, up from the previous 77.73%, while using slightly fewer parameters. The approach is well-motivated and intriguing. Given the rising prominence of LLMs, this paper could be of significant value to the community. We strongly encourage the authors to address and refine the areas highlighted by the reviewers.

---

### Decision · Program_Chairs · 2023-10-07

**Decision:**

Accept-Findings

**Comment:**

The paper introduces GEEK, a method that employs multi-step reasoning and retrieval from external knowledge sources to answer implicit questions. Such questions typically pose challenges for LLMs due to outdated parametric knowledge. GEEK incorporates both question decomposition and a halting condition. On StrategyQA, GEEK achieves a new state of the art of 78.17%, up from the previous 77.73%, while using slightly fewer parameters. The approach is well-motivated and intriguing. Given the rising prominence of LLMs, this paper could be of significant value to the community. We strongly encourage the authors to address and refine the areas highlighted by the reviewers.